# SOAR-RL: Safe and Open-Space Aware Reinforcement Learning for Mobile Robot Navigation in Narrow Spaces

**DOI:** 10.3390/s25175236

**Published:** 2025-08-22

**Authors:** Minkyung Jun, Piljae Park, Hoeryong Jung

**Affiliations:** 1Department of Mechanical Engineering, Konkuk University, 120 Neungdong-ro, Gwangjin-gu, Seoul 05029, Republic of Korea; minkyung2628@konkuk.ac.kr; 2AI SoC Department, Electronics and Telecommunications Research Institute (ETRI), 218 Gajeong-ro, Yuseong-gu, Daejeon 34129, Republic of Korea; pjpark@etri.re.kr

**Keywords:** deep reinforcement learning, mobile robot navigation, narrow space environments (NSEs), sector-based spatial representation, socially aware planning

## Abstract

As human–robot shared service environments become increasingly common, autonomous navigation in narrow space environments (NSEs), such as indoor corridors and crosswalks, becomes challenging. Mobile robots must go beyond reactive collision avoidance and interpret surrounding risks to proactively select safer routes in dynamic and spatially constrained environments. This study proposes a deep reinforcement learning (DRL)-based navigation framework that enables mobile robots to interact with pedestrians while identifying and traversing open and safe spaces. The framework fuses 3D LiDAR and RGB camera data to recognize individual pedestrians and estimate their position and velocity in real time. Based on this, a human-aware occupancy map (HAOM) is constructed, combining both static obstacles and dynamic risk zones, and used as the input state for DRL. To promote proactive and safe navigation behaviors, we design a state representation and reward structure that guide the robot toward less risky areas, overcoming the limitations of traditional approaches. The proposed method is validated through a series of simulation experiments, including straight, L-shaped, and cross-shaped layouts, designed to reflect typical narrow space environments. Various dynamic obstacle scenarios were incorporated during both training and evaluation. The results demonstrate that the proposed approach significantly improves navigation success rates and reduces collision incidents compared to conventional navigation planners across diverse NSE conditions.

## 1. Introduction

Mobile robots are increasingly being deployed across a range of service domains, such as last-mile delivery and security patrols. Particularly, the widespread adoption of door-to-door delivery services has led to more frequent robotic navigation in outdoor pedestrian areas, such as sidewalks and crosswalks [1], and indoor environments, such as building corridors [2], with dense human traffic. As robots begin to coexist with humans within constrained areas, known as narrow-space environments (NSEs), ensuring both safety and social acceptability in autonomous navigation has become a critical challenge.

In constrained and dynamic spaces [3,4], robots frequently operate in close proximity to humans, where traditional reactive collision avoidance is insufficient to achieve smooth and natural navigation. Instead, robots must be capable of recognizing and predicting pedestrian movements in real time [5,6] while comprehending the surrounding spatial structure to generate behaviors that are both safe and socially coherent [7,8,9,10].

Conventional path planning frameworks typically rely on static maps of predefined environments, combining global planners such as A* [11] or Dijkstra’s algorithm [12] with local reactive methods such as the dynamic window approach (DWA) [13] or timed elastic band (TEB) [14]. Although these methods perform well in static environments, they fall short when dealing with newly detected or dynamic obstacles [15]. Consequently, recent studies have emphasized socially-aware navigation where robots adapt their paths in anticipation of human behavior and implicit norms [16,17,18].

To address this need, recent studies have explored deep reinforcement learning (DRL) and multi-sensor fusion. The fusion of 3D LiDAR and RGB camera data enables accurate and real-time pedestrian detection and tracking [19,20], whereas human trajectory prediction facilitates dynamic risk estimation and yields informed navigation decisions [5,6,21,22,23]. Several studies introduced the concepts of dynamic danger zones [24,25], risk maps, and socially informed state encodings [26,27] to enhance robot decision making in crowded environments. Although these advances surpass traditional planning approaches, several methods primarily focus on collision avoidance and often neglect the importance of identifying and leveraging relatively safe or open regions during navigation.

To overcome these challenges, we propose a reinforcement learning (RL) framework for NSEs, where robots and pedestrians move in close proximity. In contrast to traditional approaches that focus solely on collision avoidance, the proposed method enables robots to identify and utilize open spaces for navigation. The framework fuses 3D LiDAR and RGB camera data to detect pedestrians in real time and constructs a risk-aware map based on perceived information. This map is then used to extract spatial obstacle features that serve as inputs for robot learning. Consequently, the robot can safely navigate confined NSEs by considering obstacle avoidance and traversable space selection. The key contributions of this study are summarized as follows.

Sensor fusion-based risk-aware map generation: We implement a pedestrian-aware perception system by fusing 3D LiDAR and RGB camera inputs to individually detect pedestrians and track their positions, velocities, and directions. This information is used to define personalized danger zones [24] and construct a human-aware occupancy map (HAOM) by integrating dynamic risk with static obstacles.Sector-based spatial feature extraction: Building on the generated HAOM, we propose sector-based spatial encoding that extracts obstacle proximity, pedestrian velocity, and available free space as relative spatial features between the robot and surrounding objects. These features provide structured spatial information that allows the robot to effectively perceive and comprehend its surrounding environment during navigation.RL for open-space-seeking behavior: To encourage the robot to actively seek open and safe directions, we incorporate the open-space preference into the RL framework by designing reward functions that go beyond simple collision avoidance. The reward combines multiple components, including goal proximity, dynamic risk avoidance, and open-space alignment, enabling the robot to learn safer and more traversable paths, even in complex and crowded environments.

## 2. Materials and Methods

### 2.1. System Overview

This study proposes an integrated path planning system for NSEs that enables safe and efficient navigation of mobile robots near dynamic pedestrians. As shown in Figure 1, the overall framework combines (a) sensor fusion for pedestrian detection; (b) HAOM generation, integrating static and dynamic obstacles; and (c) RL-based path planning using encoded spatial information.

The 3D LiDAR point cloud is pre-processed to extract points within the robot’s forward field of view (FOV) and fused with pedestrian bounding boxes detected from the RGB camera images. This multimodal fusion enables the identification and tracking of individual pedestrians and the estimation of their positions, velocities, and heading directions in real time. Based on these estimates, personalized danger zones [24] are assigned for each pedestrian. Combined with static obstacles, this information is used to generate an HAOM that reflects both spatial constraints and dynamic risks posed by pedestrians. The HAOM is then converted into a structured state vector containing obstacle proximity, pedestrian dynamics, and open-space information, which serves as an input for the RL agent. The RL policy learns to navigate by not only avoiding collisions but also proactively selecting safer and more traversable paths based on real-time spatial awareness.

Figure 2 illustrates the system’s execution pipeline during real-world deployment. The robot detects pedestrians using onboard RGB and LiDAR sensors, whose outputs are fused to generate the HAOM. This representation, together with the robot’s current state (pose and previous action) and the designated target goal, is provided to the trained ONNX-based policy network. The policy network then produces velocity commands, which are transmitted via ROS to actuate the robot. This closed-loop process operates continuously, enabling real-time navigation in dynamic environments.

### 2.2. Individual Human Recognition via 3D LiDAR–RGB Camera Fusion

To enable safe and precise navigation in NSEs, we developed a multi-sensor fusion module to recognize individual pedestrians using 3D LiDAR and an RGB camera. As shown in Figure 3, the pipeline consists of three primary stages: (a) pre-processing of 3D LiDAR point clouds; (b) human detection through RGB–LiDAR data fusion; and (c) individual-level human clustering and tracking.

#### 2.2.1. Preprocessing of 3D LiDAR Data

Figure 3a shows the preprocessing of the 3D LiDAR point cloud data. The raw point cloud acquired from 3D LiDAR is first segmented into ground and nonground components using the patchwork algorithm [28], which exploits the local smoothness and elevation priors, and the ground points are subsequently removed. This segmentation is based on the premise that navigation-relevant obstacles, such as pedestrians or indoor furniture, are primarily located within nonground regions. The region of interest (ROI) is restricted to the robot’s forward-facing zone to facilitate spatial alignment with the FOV of the RGB camera. The extracted point clouds are down sampled to ensure computational efficiency.

#### 2.2.2. RGB–LiDAR Fusion for Individual Human Detection

Pedestrian detection is performed on the RGB image using a YOLO-based object detector that outputs bounding boxes for each detected person. In our implementation, we used the YOLOv12 small model to ensure real-time performance. The detection threshold is set with confidence ≥ 0.5, and the detection module is configured to maintain the identity of tracked pedestrians across frames while restricting recognition to the person class only. For each bounding box, a unique ID is assigned to its center point. The ROI-filtered LiDAR point cloud obtained from Section 2.2.1 is then projected onto the image plane, aligning the 3D points with the 2D bounding boxes. Each projected point is matched to the nearest bounding box by computing its Euclidean distance to the box center and is assigned to the corresponding group of that bounding box.

Because the original LiDAR points are 3D vectors, but projection onto the 2D image removes depth information, points belonging to non-pedestrian objects behind the pedestrian (e.g., walls) can get incorrectly assigned to the group of pedestrians. The Euclidean distances from the robot to all the points in each group are computed to eliminate distant and potentially irrelevant points in the pedestrian point cloud. The minimum of these distances is defined as min_range, and an experimentally determined margin, $\delta_{margin}$, is added to define a threshold. Only points within $\mathrm{min\_range}+\delta_{margin}$ are retained, whereas those beyond this threshold are considered outliers and removed. This filtering strategy effectively excludes outliers as illustrated in Figure 4.

After filtering the irrelevant points, each group retains its initially assigned unique ID. To visualize the clustering results, each pedestrian cluster is assigned a distinct color corresponding to its ID.

#### 2.2.3. Human Tracking and Dynamic Risk Estimation

The centroid is computed and tracked for each identified pedestrian cluster. Based on the centroid positions $p_{i}$, velocity vectors $v_{i}$ are estimated by calculating the temporal differences across consecutive frames. These vectors represent both the heading direction and walking speed of each pedestrian. Using this information, individual-specific dynamic danger zones [24] are generated, enabling localized risk estimation around moving agents. These zones dynamically adapt to pedestrian movement and are critical for modeling human-aware spatial constraints. The danger zone for each pedestrian is defined by Equations (1) and (2) as follows:(1)$$r_{i}\;=\;m_{v}v_{i}+r_{static},$$(2)$$\theta_{i}=\frac{11\pi }{6}e^{-1.4v_{i}}+\frac{\pi }{6},$$
where $r_{i}$ denotes the sector radius, and $\theta_{i}$ denotes the sector angle of i-th pedestrian, both determined based on the pedestrian’s walking velocity $v_{i}$. The sector radius $r_{i}$ increases linearly with speed, where $r_{static}$ represents the stride length of a stationary person, and $m_{v}$ is an empirically derived scaling factor. The sector angle $\theta_{i}$ is exponentially adjusted according to $v_{i}$ to reflect narrower danger zones at higher speeds, as described in Equation (2). This allows the danger zone to adaptively reflect pedestrian motion and risk levels. Thus, each pedestrian is individually recognized, and their position, velocity, and corresponding danger zone can be extracted in real time based on cluster centroids. The visual representation of this formulation, including the radius $r_{i}$ and sector angle $\theta_{i}$, is illustrated in Figure 3c.

### 2.3. HAOM Generation

To generate the HAOM, the ROI-filtered point cloud is segmented into static and dynamic components. Points that do not overlap with any detected pedestrian clusters are classified as static obstacles, representing structures such as walls or fixtures. For the dynamic components, the centroid positions of pedestrian clusters $p_{i}$ and their corresponding danger zones (defined in Section 2.2.3) are considered. Using this information, the HAOM is constructed as a multilayered occupancy map, as illustrated in Figure 5.

${Layer}_{Static}$: Represents static obstacles identified from the point cloud;${Layer}_{Human}$: Contains the centroid positions of the detected pedestrian clusters;${Layer}_{Danger\;Zone}$: Encodes individualized danger zones around each pedestrian.

The regions, which are not covered by the static, human, or danger zone layers, constitute the open-space region, implicitly capturing the remaining traversable area around the robot.

The final HAOM encodes both the spatial geometry and dynamic pedestrian-induced risk, enabling spatial reasoning for safe navigation. The map is generated with a resolution of 0.01 m, and the grid consists of 1000 × 1000 pixels covering a 10 m × 10 m region, corresponding to the robot’s forward FOV.

### 2.4. Design of the Reinforcement Learning System for Open-Space-Seeking Behavior

#### 2.4.1. State Definition

We designed a structured and multidimensional state vector for safe and spatially aware navigation in NSEs, as described in Equation (3):(3)$$S_{t}=\left[s_{t}^{d},\;s_{t}^{v},s_{t}^{o},p_{t}^{r},p_{t}^{g},a_{t-1}\right]\;,$$
where $S_{t}$ denotes the state vector at time *t*. The HAOM integrates static obstacles, centroid positions of pedestrian clusters, personalized danger zones, and open spaces. The HAOM is partitioned into uniformly distributed sectors, and spatial features are extracted for each sector. The distance calculation between the robot and obstacles is defined in Equation (4):(4)$$s_{t}^{d}=\left[d_{1},d_{2}\ldots ,d_{n}\right],\;d_{i}=\frac{r_{obs,i}}{r_{max}}\;,$$
where $r_{obs,i}$ denotes the distance between the robot and nearest static obstacle or pedestrian cluster centroid within sector $q_{i}$, $r_{max}$ represents the half-length of the HAOM, and $d_{i}\;=\;\frac{r_{obs,i}}{r_{max}}$ is the normalized distance used as the distance feature. In Figure 6, the front area of the robot is evenly divided into angular sectors, and each divided region is denoted by sector $q_{i}$. Within each sector $q_{i}$, the Euclidean distance between the robot’s position and the closest pixel, labeled as either ${Layer}_{Static}$ or ${Layer}_{Human}$, is computed. The minimum distance $r_{obs,i}$ is normalized by $r_{max}$, yielding the normalized distance feature $d_{i}\in \left[0,1\right]$ used in the state vector. If no obstacle is detected within the sector, $d_{i}$ is set to 0.

The velocity vector feature is defined by Equation (5):(5)$$s_{t}^{v}=\left[v_{1},v_{2}\ldots ,v_{m}\right],\;$$
where $v_{i}$ represents the projected velocity vector of a pedestrian within sector $q_{i}$, with components $v_{x,i}\;and\;v_{y,i}$ denoting the velocities along the x- and y-axes, respectively. The area within the camera’s FOV is uniformly divided into *m* angular sectors, each denoted as $q_{i}$, as shown in Figure 7. The individually identified pedestrians are tracked to estimate their velocity vectors using the perception system described in Section 2.2.3. For each tracked pedestrian, the tracking marker $p_{i}$ is projected onto one of the sectors $q_{i}$, and the corresponding velocity vector $v_{i}$, computed from the displacement over time, is assigned to that sector. Furthermore, since the robot is also in motion, the velocity is represented as the relative velocity between each pedestrian and the robot. When no dynamic obstacle is present in a sector, the corresponding velocity value is set to zero. If multiple pedestrians fall into the same sector, the velocity of the pedestrian closest to the robot is selected.

The traversable distance information $s_{t}^{o}$ is defined in Equation (6):(6)$$s_{t}^{o}=\left[o_{1},o_{2}\ldots ,o_{m}\right],\;o_{i}=\frac{r_{obs,i}^{*}}{r_{max}},$$
where $r_{obs,i}^{*}$ denotes the maximum obstacle-free traversable distance along sector $q_{i}$, $r_{max}$ represents the half-length of the HAOM and $o_{i}=\;\frac{r_{obs,i}^{*}}{r_{max}}$ is the normalized open-space value used as the open-space feature. Open-space features are computed by conservatively accounting for static and dynamic obstacles using danger zones. As shown in Figure 8, within each sector $q_{i}$, the traversable distance $r_{obs,i}^{*}$ is calculated along the sector direction by excluding all obstacles while including static elements and dynamic agents, along with their respective danger zones. This value is normalized by $r_{max}$ to produce the open-space feature $o_{i}\in \left[0,\;1\right]$. Unlike $d_{i}$ as in Equation (4), which indicates the proximity to an obstacle and becomes zero when no obstacle is detected, $o_{i}$ quantifies the amount of free space in a sector and reaches 1 when the sector is entirely free of obstacles.

In addition to the sector-based features, the state vector includes the robot’s current position, goal position, and action at the previous time step to ensure smoother behavior and temporal consistency.(7)$$p_{t}^{r}=\;\left[x_{r},y_{r},\;z_{r}\right],$$(8)$$p_{t}^{g}=\;\left[x_{g},y_{g},\;z_{g}\right],$$(9)$$a_{t-1}=\;\left[v_{t-1},\omega_{t-1}\right],$$
where $p_{t}^{r}\;$, $p_{t}^{g}$, and $a_{t-1}$ represents the robot’s current position, goal position, and the previous action at timestep t−1, respectively. By combining sector-based observations with these additional variables, the state vectors capture the static structures, dynamic agent behaviors, and temporal information. This structured state representation supports socially aware decision making and robust path planning in crowded environments.

#### 2.4.2. Reward Design

We define a reward function consisting of seven distinct components to enable the robot to avoid collisions and proactively seek navigable open spaces in NSEs. These components are designed to complement each other and are combined as shown in Equation (10):(10)$$R_{t}=\omega_{1}\cdot R_{a}+\omega_{2}\cdot R_{h}+\omega_{3}\cdot R_{p}+\omega_{4}\cdot R_{\mathrm{dz}}+\omega_{5}\cdot R_{d}+\omega_{6}\cdot R_{o}+\omega_{7}\cdot R_{c},$$
where the total reward $R_{t}$ is computed as a weighted sum of the individual components, with each term multiplied by its corresponding weight $\omega_{i}$. The weights are manually tuned through iterative experiments. The reward components are listed in Table 1 and formulated as follows:(11)$$R_{a}=\left\{\begin{array}{cc} \;+1.0 & \;if\;d_{g}\leq \;d_{th}\\ 0.0 & otherwise \end{array}\right.,$$(12)$$R_{h}=\left\{\begin{array}{cc} 1.0 & if\;cos\theta_{g}>\theta_{th}\\ cos\theta_{g} & otherwise \end{array}\right.,\;\;(cos\theta_{g}=\frac{h_{r}\cdot h_{g}}{\left\|h_{r}\right\|\cdot \left\|h_{g}\right\|}),$$(13)$$R_{p}=\frac{-d_{g,t}+d_{g,t-1}}{∆t},$$(14)$$R_{dz}=\left\{\begin{array}{cc} \;0.1-0.1\cdot \frac{min(d_{o},r_{dz})}{r_{dz}} & \;if\;robot\;in\;danger\;zone\\ 0.0 & otherwise \end{array}\right.,$$(15)$$R_{d}=-\alpha \cdot \sum_{i=1}^{N}{\left(1-s_{o,i}\right)}^{2},$$(16)$$R_{o}=cos\theta_{o}=\frac{h_{r}\cdot h_{o}}{\left\|h_{r}\right\|\cdot \left\|h_{o}\right\|},$$(17)$$R_{c}=\left\{\begin{array}{cc} -10.0 & \;if\;collides\\ 0.0 & otherwise \end{array}\right..$$

To determine the coefficients of each reward term, a manual tuning process was conducted through iterative testing. The tuning followed a progressive strategy according to the complexity of the environment. First, in an environment with no obstacles, the weights of $R_{a}$, $R_{h}$, and $R_{p}$ were tuned to ensure that the agent could reach the goal reliably without considering obstacle avoidance. Next, for environments with static obstacles, $R_{c}$ was introduced. However, we observed that using $R_{c}$ alone often resulted in overly conservative behavior rather than active wall avoidance. To address this, $R_{d}$ was introduced to encourage the robot to maintain a safe buffer from obstacles. At this stage, the weight of the $R_{a}$ was also increased to reinforce goal-oriented navigation. Finally, in environments that included both static and dynamic obstacles, we designed and added two additional reward terms: $R_{dz}$ and $R_{o}$. These were specifically developed to guide the agent in selecting safe and traversable paths, particularly under crowded conditions. The weights were adjusted to balance obstacle avoidance and goal-reaching performance.

#### 2.4.3. RL Configuration

The action space of the robot is defined by two continuous control variables: linear ($v_{t}$) and angular velocity ($\omega_{t}$):(18)$$a_{t}=[v_{t},\omega_{t}].$$

At each timestep, the policy network outputs a pair of variables [$v_{t},\omega_{t}$]. The linear velocity is constrained to the range [−2.0, 2.0] m/s, and the angular velocity is limited to [−1.5, 1.5] rad/s, based on the specifications of the unmanned ground vehicle Jackal from Clearpath. Each episode terminates under one of the following three conditions: the robot reaches the goal, the robot collides with an obstacle, or the number of steps exceeds a predefined limit. Training was conducted using the proximal policy optimization (PPO) algorithm [29], with the following key parameters: learning rate of 0.001, discount factor *γ* of 0.99, GAE coefficient *λ* of 0.95, clipping range of 0.2, entropy coefficient of 0.01, mini-batch size of 512, and mini-epoch number of 8. The training lasted for 1000 epochs, utilizing 512 parallel environment instances for maximal training efficiency.

## 3. Results

### 3.1. Experimental Setup

The reinforcement learning and performance evaluation of the proposed method is conducted in the simulation environment constructed using NVIDIA Isaac Sim. Owing to the high computational cost of sensor-based simulations involving real-time LiDAR and RGB streaming, a lightweight environment based on preprocessed state vectors, excluding real-time sensor streams, was used during training. The simulation was performed on a system equipped with a NVIDIA GeForce RTX 4090 GPU and an Intel Core i9-14900K CPU. The demonstration of the experimental results can be found in the Appendix A.

#### 3.1.1. Environmental Layout and Course Design

To evaluate the proposed method across varying levels of difficulty in constrained environments, three corridor configurations were implemented, as illustrated in Figure 9.

Straight course: A simple, one-directional corridor resembling a sidewalk or indoor hallway. This configuration involves minimal occlusions and low collision risks and serves as a baseline scenario.L-shaped course: A right-angled corridor introducing moderate difficulty owing to limited visibility around the corners. This poses challenges for predicting pedestrian movement and planning in occluded areas.Intersection course: A four-way junction allowing multidirectional pedestrian flow. This complex layout involves frequent dynamic interactions and high uncertainty, which makes it the most challenging scenario.

In each scenario, the starting position and goal of the robot were placed at opposite ends or junctions. The obstacles were initialized randomly, producing diverse navigation scenarios dependent on both the corridor structure and dynamic agent configuration. This setup enabled a comprehensive evaluation of the generalization and robustness of the learned navigation policy in NSEs.

#### 3.1.2. Dynamic Obstacle Modeling

The dynamic obstacles were modeled as cylinders, as shown in Figure 9. At the beginning of each episode, obstacles are randomly positioned within predefined regions, and their speeds are randomly set within the range of [0.0–1.0 m/s], reflecting the typical walking speed of pedestrians in narrow environments [30,31]. The obstacle controller ensures that obstacles remain within their designated movement boundaries, allowing the reinforcement learning policy to generalize across diverse human-like motion patterns and initial configurations.

#### 3.1.3. Evaluation Metrics

The evaluation metrics used for quantitative performance assessment are listed in Table 2.

### 3.2. Performance of the Proposed Method Under Various Environmental Conditions

To evaluate the robustness of the proposed method under diverse environmental settings, experiments were conducted by combining three scenario types (straight, L-shaped, and intersecting) with varying numbers of dynamic obstacles, ranging from one to four. Table 3 and Figure 10 summarize the performance of the proposed method under varying dynamic obstacle densities across the three scenarios.

As the number of dynamic obstacles increases from one to four, the success rate generally decreases, and the collision rate increases. The success rates ranged from 98.0% to 87.0%, whereas the collision rates ranged from 1.0% to 13.0% across all scenarios. Scenario 3 yielded the highest average returns across all tested scenarios, with a value of 225.2 ± 39.5 for a single dynamic obstacle, and a slight reduction to 209.8 ± 32.7 as the number of dynamic obstacles increased to four. Scenario 1 exhibited a decreasing trend from 157.1 ± 23.7 (1 obstacle) to 144.5 ± 40.2 (4 obstacles), whereas Scenario 2 exhibited a modest increase from 149.4 ± 30.5 to 166.9 ± 34.4 as the number of dynamic obstacles increased. Overall, although the absolute average return values differed across the scenarios, the intra-scenario variations due to the increasing number of dynamic obstacles remained moderate, generally within 10–12%.

To further assess the performance of the proposed method under more challenging conditions, additional tests were conducted in the intersection course with six dynamic obstacles, as shown in Figure 10. In this densely crowded intersection scenario, the proposed method achieved a success rate exceeding 80%.

### 3.3. Ablation Study on Open-Dir Alignment Reward ($R_{o}$)

The contribution of the open-direction alignment reward $R_{o}$ as in Equation (16) was evaluated through ablation experiments in all scenarios. As listed in Table 4, the presence of $R_{o}$ consistently enhanced both SR and CR.

The addition of open-direction alignment reward $R_{o}$ led to consistent improvements across all scenarios. In Scenario 1, the success rate increased by +14.0% (from 75.0% to 89.0%), and the collision rate decreased by 14.0% (from 25.0% to 11.0%). In Scenario 2, $r_{o}$ improved the success rate by +15.0% and reduced the collision rate by –10.0%. In Scenario 3, the success rate rose by +16.0%, and the collision rate decreased by 14.0%.

### 3.4. Performance Comparison with Traditional Navigation Methods

This section compares the proposed safe and open-space-aware RL (SOAR-RL) method with conventional ROS-based local planners. The baseline methods include three widely used planners: DWA [13], TEB [14], and trajectory rollout (TR) [32], which is a variant of DWA. These local planners have been extensively adopted in both industrial and academic domains and serve as standard benchmarks for mobile robot navigation. The experiments were conducted in a simulated environment using NVIDIA Isaac Sim integrated with ROS. All planners processed the sensor data simulated using a Velodyne VLP-16 LiDAR system. The global planner was fixed using Dijkstra’s algorithm, whereas each local planner and the proposed method were evaluated under the same maps and initial conditions, with 100 trials per scenario. Table 5 lists the quantitative analysis results.

In Scenario 1, the proposed method (SOAR-RL) achieved the highest success rate of 93.0%, significantly outperforming the TR [32] (49.0%), DWA [13] (39.0%), and TEB [14] (32.0%) methods. It also had the lowest collision rate (7.0%) and zero timeouts. In contrast, TR exhibited a 13.0% collision rate and a 38.0% timeout rate. Although as shown in Figure 11c, TR showed a path length similar to that of (d) SOAR-RL, substantially more time was required to reach the goal. DWA (Figure 11a) and TEB (Figure 11b) frequently failed early in the trajectory, often resulting in immediate collisions with nearby obstacles. These early terminations explain the shorter average path lengths and reduced navigation times of DWA and TEB compared with those of SOAR-RL.

In Scenario 2, SOAR-RL exhibited superior performance, achieving a success rate of 94.0% compared with TR (48.0%), DWA (36.0%), and TEB (35.0%). It also achieved the lowest collision rate (6.0%), with no timeouts, whereas TR experienced 31.0% timeouts and 21.0% collisions. As depicted in Figure 12, DWA and TEB frequently failed while turning around the L-shaped corner, where restricted visibility and limited space made trajectory adjustment difficult. These failures resulted in shorter path lengths and times (DWA: 3.26 m, 12.64 s; TEB: 3.89 m, 15.17 s) compared with those of SOAR-RL, which followed a longer yet successful trajectory (15.10 m, 25.27 s). The higher average speed of SOAR-RL (0.61 m/s) also demonstrates efficient movement despite extended paths.

In Scenario 3, SOAR-RL maintained a strong performance in the most complex intersection layout, with a success rate of 88.0%, outperforming TR (49.0%), TEB (41.0%), and DWA (42.0%). The collision rate was 12.0%, and timeout occurred only 4.0% of the time. As shown in Figure 13, traditional planners, such as TR and DWA, often collided midway or failed to reach the goal within the allowed time, particularly in dense zones where crossing pedestrians obstructed the path. These behaviors resulted in higher timeout rates and shorter navigation durations for most methods. In contrast, SOAR-RL achieved the longest path length (20.99 m) and navigation time (34.03 s), thereby successfully avoiding collisions and delays.

### 3.5. Performance Comparison with AI-Based Crowd Navigation Methods

This section presents a comparative analysis of the proposed method against several AI-based crowd navigation methods. The comparison includes a traditional geometric approach, ORCA [33], and state-of-the-art deep learning models, such as CADRL [34], LSTM-RL [35], and DSRNN [36]. All experiments were carried out in a 6 m × 6 m square environment populated with five dynamic obstacles. Each methodology was evaluated over 500 trials, and the quantitative results are summarized in Table 6. As presented in Table 6, the proposed SOAR-RL achieved a success rate of 94.0% and a collision rate of 6.0%. This rate is higher than those of other AI-based models, such as DSRNN (93.0%) and LSTM-RL (89.0%). The low collision rate with zero timeouts indicates that the proposed method shows stable and reliable navigation performance.

In terms of efficiency, SOAR-RL achieved the shortest average navigation time of 5.28 s with an average path length of 4.21 m. By contrast, DSRNN and LSTM-RL followed longer paths (10.01 m and 7.05 m, respectively), which reflects more conservative avoidance strategies and resulted in longer navigation times. ORCA generated a shorter path of 2.91 m, but this was associated with more aggressive maneuvering, leading to a collision rate of 68.0%. These results suggest that the proposed SOAR-RL provides an effective trade-off between safety and efficiency compared to the previous methods.

### 3.6. Real-Time Performance Analysis

To verify whether the proposed RL-based navigation framework satisfies real-time operation requirements, we evaluated its computational performance across multiple test scenarios. Table 7 presents the FPS recorded over the last 100 steps of simulation for each scenario. Scenarios 1, 2, and 3 were conducted with four dynamic obstacles in different spatial configurations, while Scenario 4 included the intersection and six dynamic obstacles. As shown in Table 6, all scenarios achieved an average FPS above 21, which corresponds to an update interval of approximately 47 ms or faster. This confirms that the RL-based navigation system can process sensor data and make decisions within the time constraints required for real-time pedestrian interaction, assuming a maximum human walking speed of 2 m/s.

## 4. Discussion

The proposed method consistently maintained high performance as environmental complexity and dynamic obstacle density increased. As listed in Table 3, the success rate declined moderately as the number of obstacles increased from one to four but remained above 87% in all scenarios. Similarly, the collision rate increased slightly but did not exceed 13%, indicating that the spatially informed state design and reward structure effectively supported reliable navigation, even in cluttered environments. In addition to the success rate, the average return was included as a metric to illustrate the consistency of learning performance. Even as the number of obstacles increased, the average return remained relatively stable, indicating robust learning. The variation in the average return across scenarios is attributed to structural differences such as the goal distance and corridor width. Notably, the highest average return was observed in Scenario 3, likely due to the extended goal path and additional rewards accumulated through strategic detours and obstacle avoidance in the more complex layout. To test the robustness of the system further, an additional experiment was conducted in an extremely dense scenario with six dynamic obstacles. As shown in Figure 10, the robot successfully reached the goal while avoiding collisions with both moving agents and static walls. Even under high-density conditions with frequent interactions, the method achieved a success rate exceeding 80%, demonstrating its strong adaptability to complex social navigation environments.

The Open-Dir alignment reward proved essential for enhancing the path planning performance. Moreover, the ablation study showed that removing this reward caused the robot to rely heavily on reactive collision avoidance, often resulting in frequent collisions and timeouts owing to a lack of directional consistency. In contrast, with the reward enabled, the robot not only avoided obstacles but also maintained a consistent heading toward the traversable open space. This led to notable performance improvements, with success rates increasing from 75% to 89% in Scenario 1, from 78% to 93% in Scenario 2, and from 75% to 91% in Scenario 3. Both collision and timeout rates were reduced in all scenarios. Although a few timeout cases remained in Scenario 3, they likely stemmed from the robot choosing conservative detours at complex intersection layouts. Additionally, training took longer when an open-direction alignment reward was applied. This is because, without the reward, episodes often ended quickly owing to early collisions, resulting in shorter episode durations. In contrast, when the reward was enabled, the robot tended to reach the goal more frequently, which naturally extended the length of each episode. Therefore, the difference in total training time for 1000 epochs reflects the fact that the robot learned to avoid collisions and successfully reach the goal only with the open-direction alignment reward. Overall, the inclusion of the open-direction alignment reward encouraged more strategic path selection and contributed to more stable and efficient policy behavior.

The proposed method (SOAR-RL) demonstrated a robust balance between safety and efficiency compared to both conventional ROS-based local planners, such as DWA [13], TEB [14], and TR [32], as well as other AI-based crowd navigation baselines, including CADRL [34], LSTM-RL [35], and DSRNN [36]. Traditional planners rely primarily on reactive control based on immediate sensor inputs. As a result, they often face limitations in anticipating the future motion of dynamic obstacles. In contrast, SOAR-RL incorporates dynamic motion information into its state representation. This design contributed to higher success rates and lower collision rates across the evaluated scenarios. The comparison with specialized crowd navigation methods shows a similar trend. ORCA, for instance, produced relatively short paths but, given its limited consideration of longer-term motion, exhibited a substantially higher collision rate. Conversely, AI-based methods such as DSRNN and LSTM-RL generally employed more conservative avoidance strategies. This led to longer navigation times and increased path lengths. The proposed method generates trajectories that achieve a more favorable balance between safety and efficiency compared to both traditional planners and previous AI-based crowd navigation approaches.

Despite the advantages of the proposed method, several limitations must be addressed. First, the reward function exhibited high sensitivity to the relative weights of its components, and minor adjustments often led to noticeable changes in performance. Because the current weights were selected through manual tuning, future work should explore automated reward optimization techniques. Second, the HAOM-based state representation relies solely on the RGB camera’s field of view, which limits the robot’s situational awareness. This constraint can lead to collisions in lateral or rear blind spots outside the camera’s coverage. Moreover, the pedestrian detection module depends on YOLO-based bounding box detection, which may occasionally fail due to occlusion, rapid pedestrian movement, or challenging lighting conditions. Such missed detections can produce incomplete or inaccurate risk zone maps, reducing the reliability of the HAOM and increasing the likelihood of unsafe path decisions. In future work, we plan to integrate a pedestrian tracking module with motion prediction to preserve awareness of nearby pedestrians even during temporary occlusions or detection failures. This enhancement will also enable smoother temporal continuity in risk zone estimation. Future studies should include unified benchmarks for the methods. Moreover, real-world robot experiments and broader scenario tests are planned to further assess the generalizability and deployment feasibility of this method.

## 5. Conclusions

This study proposed a DRL-based path planning framework for mobile robots operating in NSEs, where frequent close interactions with dynamic pedestrians present significant navigation challenges. To enable socially aware behavior in such complex and constrained settings, we developed a sensor-fusion-based perception module that integrates 3D LiDAR and RGB camera data for real-time pedestrian detection. An HAOM was constructed by combining static obstacle positions with dynamically generated danger zones [24], which reflect the estimated pedestrian motion trajectories. For effective learning, a 98-dimensional state vector was designed to represent the spatial distribution of obstacles, dynamic velocities, and traversable directions within a sector-based FOV. Additionally, a multi-objective reward function was established to promote collision avoidance and encourage the selection of safer and more open directions toward the goal.

Experimental validation in simulated environments featuring straight, L-shaped, and intersection corridors with varying numbers of dynamic obstacles confirmed the robustness of the proposed method. It consistently achieved high success rates and low collision rates across all scenarios, demonstrating reliable and adaptive path planning performance as complexity increased. Our findings demonstrate that SOAR-RL achieves robust and socially compliant navigation in NSEs, thus paving the way for real-world service deployment and academic exploration. This study contributes to the development of safe, human-aware autonomy. It provides a foundation for future extensions, including deployment in physical settings, improved sensing capabilities, and benchmarking against state-of-the-art RL-based planners.

## Figures and Tables

**Figure 1 sensors-25-05236-f001:**
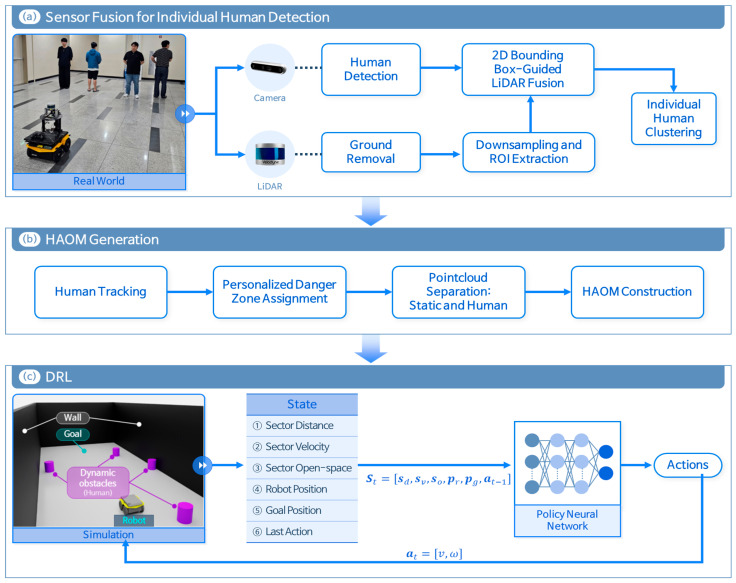
Architecture of the sensor fusion and deep reinforcement learning (DRL)-based path planning. The mobile robot used was a Clearpath Jackal, equipped with a Velodyne VLP-16 LiDAR and an Intel RealSense D435 camera, running on Ubuntu 20.04 with ROS Noetic.

**Figure 2 sensors-25-05236-f002:**
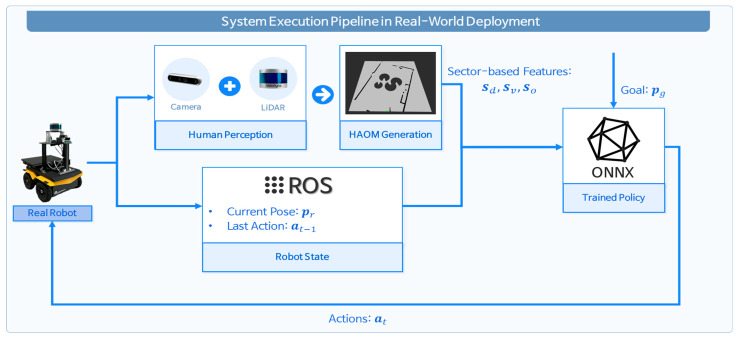
Real-world execution pipeline of the proposed method.

**Figure 3 sensors-25-05236-f003:**
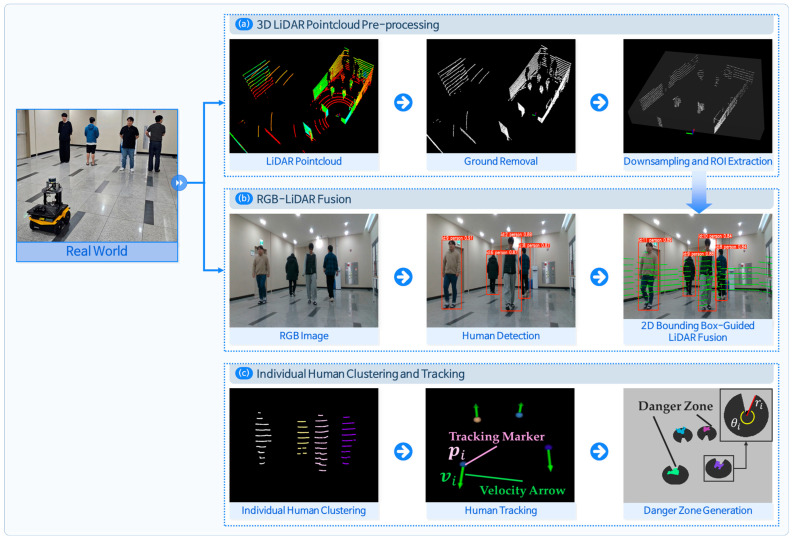
Sensor fusion pipeline for human-aware navigation. (**a**) 3D LiDAR preprocessing with ground removal and extracting region of interest (ROI) point cloud; (**b**) human detection using RGB image and bounding box fusion with LiDAR; and (**c**) individual-level human clustering with tracking.

**Figure 4 sensors-25-05236-f004:**
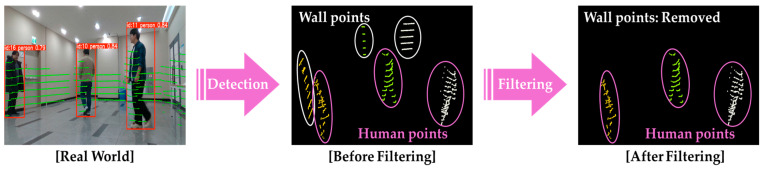
Procedure for removing non-pedestrian point clouds.

**Figure 5 sensors-25-05236-f005:**
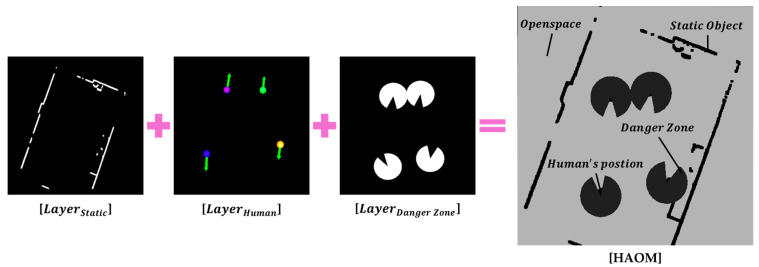
Human-aware occupancy map (HAOM) comprising static obstacles, centroid positions of pedestrian clusters, dynamic danger zones, and open space.

**Figure 6 sensors-25-05236-f006:**
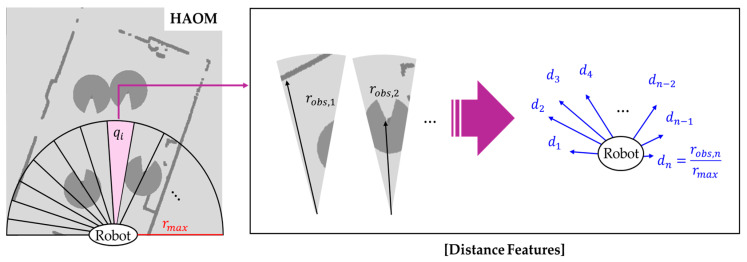
Sector-based distance feature extraction for HAOM. The robot’s front-facing area is divided into *n* uniform angular sectors $q_{i}$. For each sector, the minimum Euclidean distance $r_{obs,i}$ to static obstacles or the human clusters’ centroid is measured, normalized by the maximum sensing range $r_{max}$ and encoded as a distance feature $d_{i}\in \left[0,\;1\right]$.

**Figure 7 sensors-25-05236-f007:**
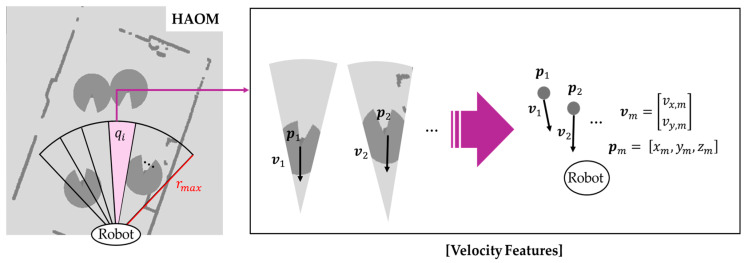
Sector-based velocity feature extraction in the camera’s field of view (FOV). The area is evenly divided into angular sectors $q_{i}$, and tracked pedestrian positions $p_{i}\;$ and velocity vectors $v_{i}$ are projected into the corresponding sectors.

**Figure 8 sensors-25-05236-f008:**
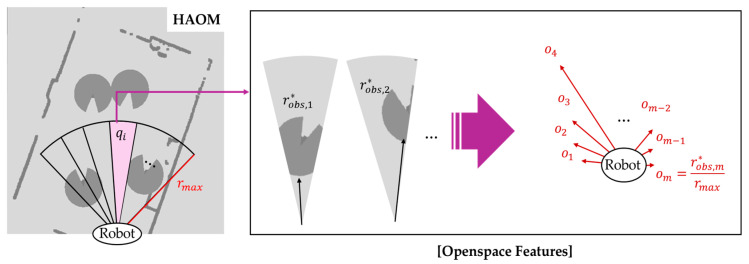
Sector-based open-space feature extraction for HAOM in the camera’s FOV. Each angular sector $q_{i}$ is analyzed to compute the normalized traversable distance $o_{i}$ considering both static and dynamic obstacles with danger zones.

**Figure 9 sensors-25-05236-f009:**
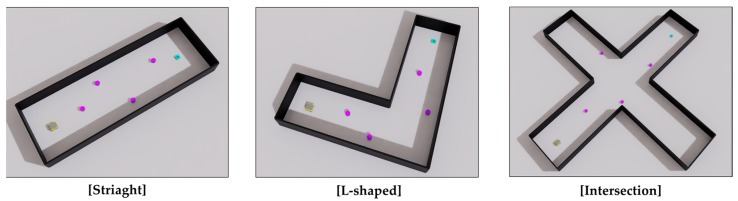
Three simulation environments used for performance evaluation. The black represents the wall structures, purple indicates dynamic obstacles, cyan denotes the goal, and gray represents the robot.

**Figure 10 sensors-25-05236-f010:**
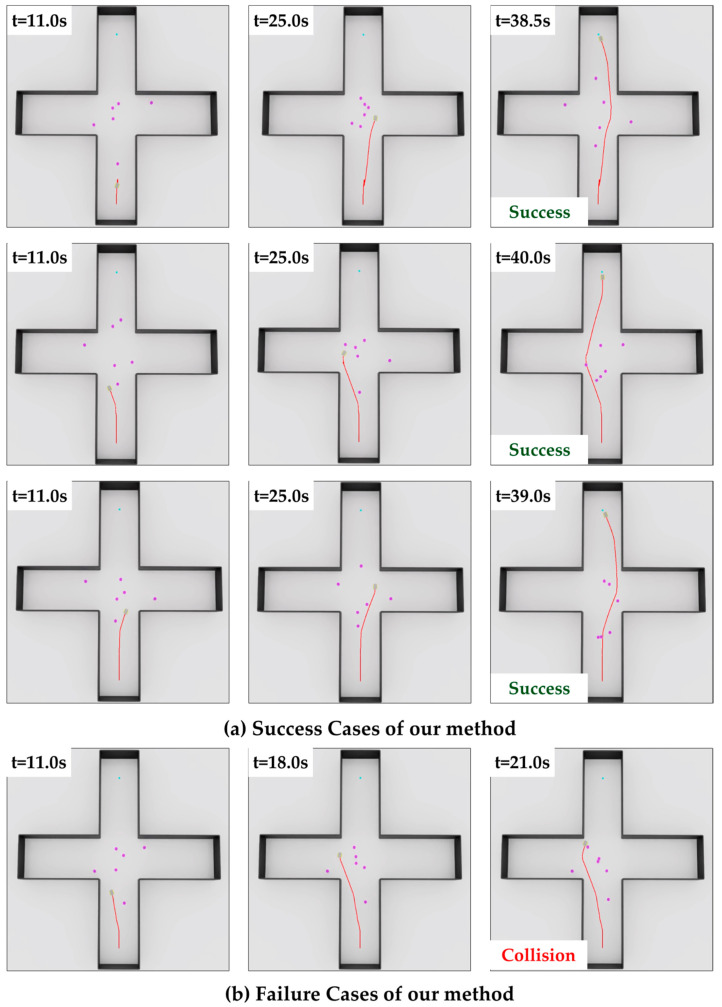
Visualizations of our method in extreme scenarios: (**a**) success cases; (**b**) failure cases. The red line represents the robot’s path.

**Figure 11 sensors-25-05236-f011:**
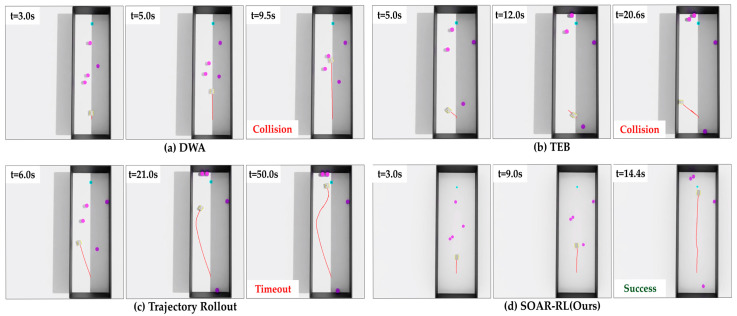
Comparison of navigation trajectories in Scenario 1: (**a**) dynamic window approach (DWA), (**b**) timed elastic band (TEB), (**c**) trajectory rollout (TR), (**d**) safe and open-space-aware RL (SAOR-RL; ours). The red line represents the robot’s path.

**Figure 12 sensors-25-05236-f012:**
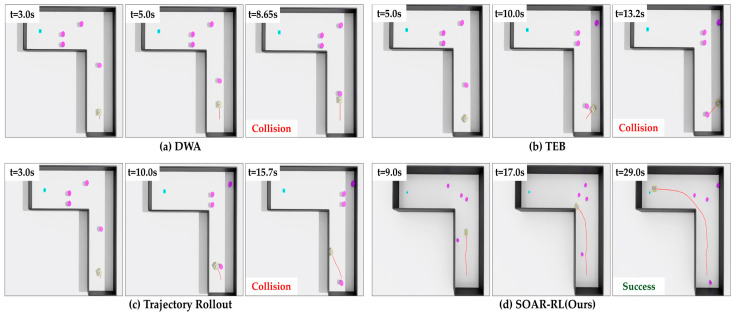
Comparison of navigation trajectories in Scenario 2: (**a**) dynamic window approach (DWA), (**b**) timed elastic band (TEB), (**c**) trajectory rollout (TR), (**d**) safe and open-space-aware RL (SAOR-RL; ours). The red line represents the robot’s path.

**Figure 13 sensors-25-05236-f013:**
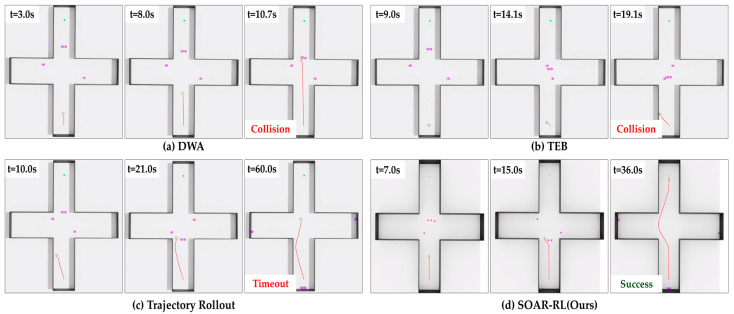
Comparison of navigation trajectories in Scenario 3: (**a**) dynamic window approach (DWA), (**b**) timed elastic band (TEB), (**c**) trajectory rollout (TR), (**d**) safe and open-space-aware RL (SAOR-RL; ours). The red line represents the robot’s path.

**Table 1 sensors-25-05236-t001:** Description of reward functions.

Reward	Explanation
$R_{a}$	Arrive: If the robot goal distance $d_{g}$ is less than $d_{th}$, gets +1.0.
$R_{h}$	Heading to goal: Measures the cosine similarity $cos\theta_{g}$ between the robot’s current direction $h_{r}$ and the direction to the goal $h_{g}$. If $cos\theta_{g}>\theta_{th}$, gets + 1.0.
$R_{p}$	Progress: Calculates the distance reduction to the goal between time steps.
$R_{dz}$	Danger zone penalty: Applies a linearly increasing penalty when the robot enters a human’s danger zone [24]. $d_{o}$ is the robot–obstacles distance; $r_{dz}$ is the danger zone radius.
$R_{d}$	Distance penalty: Penalizes proximity to static obstacles and human danger zones to reduce collision risk. $s_{o,i}$ is the open space length in each sector $q_{i}$. α is constant.
$R_{o}$	Open-Dir alignment: Computes the cosine similarity $cos\theta_{o}$ between the robot’s current direction $h_{r}$ and the central direction vector of the widest open space $h_{o}$.
$R_{c}$	Collision penalty: Applies upon collision with obstacles.

**Table 2 sensors-25-05236-t002:** Evaluation metrics used for performance assessment.

Metrics	Explanation
SR	Success rate: Percentage of episodes in which the robot successfully reaches the goal without collisions or timeouts.
CR	Collision rate: Percentage of episodes in which the robot collides with any obstacle.
TOR	Timeout rate: Percentage of episodes in which the robot fails to reach the goal within the time limit.
μR ± σR	Cumulative reward: Mean and standard deviation of cumulative rewards across episodes, indicating overall policy performance.
μPL	Path length: Average distance (in meters) the robot travels to reach the goal.
μS	Average speed: Average velocity of the robot during successful navigation.
NT	Navigation time: Time (in seconds) taken per episode, regardless of whether the robot succeeds, collides, or times out.

**Table 3 sensors-25-05236-t003:** Performance of the proposed method in various environments.

	Scenario 1		Scenario 2		Scenario 3	
# of Dynamic Obstacles	SR (%)	μR ± σR	SR (%)	μR ± σR	SR (%)	μR ± σR
1	98.0	157.07 ± 23.68	99.0	149.39 ± 30.53	96.0	225.24 ± 39.49
2	95.0	145.82 ± 17.89	94.0	163.76 ± 33.90	95.0	215.11 ± 40.43
3	91.0	147.38 ± 32.53	88.0	164.84 ± 38.96	89.0	193.51 ± 32.33
4	87.0	144.48 ± 40.17	89.0	166.86 ± 34.36	89.0	209.77 ± 32.69

**Table 4 sensors-25-05236-t004:** Quantitative comparison with and without Open-Dir alignment reward $R_{o}$ in three scenarios with four dynamic obstacles. Upward ↑ and downward ↓ arrows indicate values that are higher and lower than the value w/o $R_{o}$.

Scenario	Method	SR (%) ↑	CR (%) ↓	TOR (%) ↓	μR ± σR ↑
Scenario 1	$w/o\;R_{o}$	75.0	25.0	0.0	54.51 ± 11.26
	w/$R_{o}$	89.0	11.0	0.0	144.48 ± 40.17
Scenario 2	$w/o\;R_{o}$	78.0	17.0	5.0	99.96 ± 25.99
	w/$R_{o}$	93.0	7.0	0.0	166.86 ± 34.36
Scenario 3	$w/o\;R_{o}$	75.0	19.0	6.0	138.40 ± 27.41
	w/$R_{o}$	91.0	5.0	4.0	209.77 ± 32.69

**Table 5 sensors-25-05236-t005:** Quantitative performance comparison of the proposed method and traditional navigation methods with four dynamic obstacles. Upward ↑ and downward ↓ arrows indicate values that are higher and lower than the result of traditional navigation methods.

Scenario	Method	SR (%) ↑	CR (%) ↓	TOR (%) ↓	μPL (m) ↑	μS (m/s) ↑	NT (s)
Scenario 1	DWA [13]	39.0	45.0	16.0	4.92	0.48	10.26
	TEB [14]	32.0	48.0	20.0	4.42	0.46	11.83
	TR [32]	49.0	13.0	38.0	8.38	0.34	22.73
	SOAR-RL (Ours)	93.0	7.0	0.0	9.43	0.66	14.39
Scenario 2	DWA [13]	36.0	48.0	16.0	3.26	0.38	12.64
	TEB [14]	35.0	47.0	18.0	3.89	0.41	15.17
	TR [32]	48.0	21.0	31.0	4.63	0.38	30.78
	SOAR-RL (Ours)	94.0	6.0	0.0	15.10	0.61	25.27
Scenario 3	DWA [13]	42.0	45.0	13.0	6.54	0.41	14.24
	TEB [14]	41.0	33.0	26.0	4.51	0.46	21.87
	TR [32]	49.0	14.0	37.0	9.88	0.37	37.57
	SOAR-RL (Ours)	88.0	12.0	0.0	20.99	0.61	34.03

**Table 6 sensors-25-05236-t006:** Quantitative performance comparison of the proposed method and crowd navigation methods with five dynamic obstacles. Upward ↑ and downward ↓ arrows indicate values that are higher and lower than the result of AI-based crowd navigation methods.

Method	SR (%) ↑	CR (%) ↓	TOR (%) ↓	μPL (m) ↓	μS (m/s)	NT (s) ↓
ORCA [33]	31.0	68.0	1.0	2.91	0.35	8.41
CADRL [34]	78.0	16.0	6.0	4.49	0.56	8.04
LSTM-RL [35]	89.0	8.0	3.0	7.05	0.87	8.10
DSRNN [36]	93.0	7.0	0.0	10.01	1.02	9.78
SOAR-RL (Ours)	94.0	6.0	0.0	4.21	0.61	5.28

**Table 7 sensors-25-05236-t007:** Runtime performance summary across different navigation scenarios.

Scenarios	Avg. FPS
Scenario 1	24.03
Scenario 2	24.53
Scenario 3	24.24
Scenario 4	21.19

## Data Availability

The data presented in this study are available on request from the corresponding author. The data are not publicly available due to institutional restrictions.

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
