# Peer review of "SOAR-RL: Safe and Open-Space Aware Reinforcement Learning for Mobile Robot Navigation in Narrow Spaces"

_sensors, 2025, doi:10.3390/s25175236_

Round 1

Reviewer 1 Report

Comments and Suggestions for Authors

The study presents an approach to enhance the safe navigation of mobile robots in narrow and complex environments. Through the effective combination of environmental perception, spatial encoding, and reinforcement learning, the authors have built a system that merges 3D LiDAR sensors and RGB cameras to detect and track pedestrians in real time, thereby creating a personalized pedestrian awareness map of the danger zone by integrating both dynamic risk factors and static obstacles.

The study also proposed a regional spatial encoding method, extracting important spatial features such as distance to objects, pedestrian velocity, and environment emptiness, helping the robot to build a structured and situation-specific spatial perception.

Scenarios of space environment, number of obstacles, various parameter values to verify the method. However, the paper would be more complete if some of the following aspects were clarified as follows.

  1. 3D Lidar is used to build a map of the operating environment (actually working with static objects) and the RGB Camera will detect the dynamic object which is the pedestrian. So in the case of many pedestrians and the occlusion of objects (due to the camera angle or the movement of pedestrians) or the space has additional static obstacles (after updating the pre-processed map with 3D Lidar), what is the solution? The application of the YOLO model to detect objects should be supplemented with a description in the main part of the paper (section 2.2.2 has been described but very briefly).
  2. With a maximum walking speed of up to 2m/s. The author can explain more about the calculations and estimates for RL to be able to process and navigate the robot in real time (data processing speed and response speed of robot actuators).
  3. The Omega i (i=1, 2,…7) of the reward function are manually selected by the authors. However, it is still necessary to describe more about the criteria or direction of selection or specific goals to get the appropriate set of weights in the adjustment process as the authors have done.
  4. The diagrams and tables in the paper are built in detail. However, there should be an overall diagram of the solution so that the method is recognized and understood more completely and intuitively.

Reviewer 2 Report

Comments and Suggestions for Authors
  1. Some abbreviations happen twice, for example, DAL and HAOM appear in both the Abstract and Introduction parts. The author may need to check all manuscript.
  2. In the contributions, the authors mentioned that the velocity of pedestrians is extracted based on the perception system. However, the author also said that the sector-based spatial encoding can extract the pedestrian velocity. It’s better to make it clear.
  3. In equations 1 and 2, it’s easy to understand the radius of the danger zone. However, the definition of sector angle is not clear. I suggest adding a figure to illustrate the definition of the danger zone. Or the illustration can be added as a sub-figure in Figure 4.
  4. As for Equation 2. I am confused about the parameter values. Why do you choose those parameter values, such as 11*pi/6 and pi/6. Those numbers seem to be defined manually.
  5. In the state definition, the velocity of an individual pedestrian is estimated as a state. However, how to estimate the velocity is not mentioned. Also, is it the absolute velocity in world coordinate or the relative velocity corresponding to the robot coordinate? Because the robot will move during obstacle avoidance.
  6. In line 230, it should be “R_{max}”, the “x” is missing.
  7. In Section 2.4.1. How did the author deal with the situation if there is no pedestrian or obstacle inside the sector? The corresponding state for the sector is regarded as zero or one?
  8. For equation 11, the author should also list the value of Ra when dg > d_{th}.
  9. In the experiment, it’s good to see the ablation experiment to verify the effectiveness of the proposed reward function. However, only the Open-Dir alignment reward is discussed in the experiment. Why doesn't the author do the ablation experiment for other reward items?
  10. Additionally, the comparison experiments are compared with traditional navigation methods. It would be convincing if it could be compared with other RL-based methods.

Reviewer 3 Report

Comments and Suggestions for Authors

The paper introduces a novel framework that enables mobile robots to safely navigate in narrow spaces. The system detects and proactively tracks pedestrians in real time to construct risk-aware map. While traditional approaches focus primarily on collision avoidance, the proposed method identifies open spaces and uses them for navigation.

Overall remarks: The paper is well-written, with clear and accessible language and a coherent structure. The methodology is presented logically, and the technical content is easy to follow.

References: Most of the sources are recent (within the last 5 years), however one publication is dated back nearly 60 years. This raises the question, whether well-established algorithms (e.g., A*, Dijkstra) still require citation, or whether they can be regarded as common knowledge.

Literature review and contribution: The literature review is thorough and well-articulated. The contribution of the paper is clearly defined and positioned within the context of existing research.

Experiment and results: The Authors acknowledge certain simplifications in the experimental setup, such as the use of a simulated environment without real-time sensor streaming. While this is justified in the paper, it raises concerns about the method’s applicability in real-world conditions. It would be beneficial to address the computational complexity, especially in the context of real-time deployment.

Round 2

Reviewer 1 Report

Comments and Suggestions for Authors

The revised version shows improvement. 
However, the comparison with traditional methods ( 3.4. Performance Comparison with Traditional Navigation Methods) does not make sense since this is the AI based method. There exist many AI based methods. The authors should compare with the methods and declare that their methods overcome the existing ones.
